# The Link between Abdominal Obesity Indices and the Progression of Liver Fibrosis: Insights from a Population-Based Study

**DOI:** 10.3390/nu16111586

**Published:** 2024-05-23

**Authors:** María Teresa Julián, Ingrid Arteaga, Pere Torán-Monserrat, Guillem Pera, Alejandra Pérez-Montes de Oca, Irene Ruiz-Rojano, Elena Casademunt-Gras, Carla Chacón, Nuria Alonso

**Affiliations:** 1Department of Endocrinology and Nutrition, Hospital Germans Trias i Pujol, 08916 Badalona, Barcelona, Spain; mtjulian.germanstrias@gencat.cat (M.T.J.); alec148@gmail.com (A.P.-M.d.O.); elenacg95@gmail.com (E.C.-G.); 2Unitat de Suport a la Recerca (USR) Metropolitana Nord, Fundació Institut Universitari d’Investigació en Atenció Primària Jordi Gol i Gurina (IDIAP Jordi Gol), 08303 Mataró, Barcelona, Spain; iarteaga@gencat.cat (I.A.); gpera@idiapjgol.info (G.P.); ireneruro@gmail.com (I.R.-R.); cchaconv.mn.ics@gencat.cat (C.C.); 3Grup de Recerca en Malalties Hepàtiques a l’Atenció Primària (GRemHAp), IDIAP Jordi Gol, USR Metro-Nord, 08303 Mataró, Barcelona, Spain; 4Primary Healthcare Center Vall del Tenes, Gerència d’Àmbit d’Atenció Primària Metropolitana Nord, Institut Català de la Salut, 08186 Llicà d’Amunt, Barcelona, Spain; 5Germans Trias i Pujol Research Institute (IGTP), 08916 Badalona, Barcelona, Spain; 6Primary Healthcare Center Dr. Barraquer, Gerència d’Àmbit d’Atenció Primària Metropolitana Nord, Institut Català de la Salut, 08930 Sant Adrià del Besos, Barcelona, Spain; 7PhD Programme in Medicine and Translational Research, Faculty of Medicine, University of Barcelona, 08007 Barcelona, Spain; 8Department of Medicine, Universitat Autònoma de Barcelona, 08193 Barcelona, Spain; 9Center for Biomedical Research on Diabetes and Associated Metabolic Diseases (CIBERDEM), Instituto de Salud Carlos III, 28029 Madrid, Spain

**Keywords:** metabolic dysfunction-associated steatotic liver disease (MASLD), liver fibrosis, transient elastography (TE), visceral adiposity index (VAI), abdominal obesity, body mass index (BMI), waist circumference (WC)

## Abstract

There is currently no available information on the correlation between abdominal obesity indices and the risk of liver fibrosis progression. We aimed to investigate the relationship between the body mass index (BMI), waist circumference (WC), and the visceral adiposity index (VAI) with the progression of liver fibrosis. The study also evaluated the association between these indices and the prevalence of metabolic dysfunction-associated steatotic liver disease (MASLD) and liver fibrosis. A total of 1403 subjects participated in the cross-sectional and longitudinal population-based study. Liver stiffness was assessed via transient elastography, at baseline and follow-up (median: 4.2 years). The subgroup with dysglycemia was also analyzed. In the cross-sectional study, the highest quartile of VAI, BMI ≥ 30 kg/m^2^, and abdominal obesity showed significant associations with the prevalence of MASLD and liver fibrosis, as well as with fibrosis progression. However, VAI showed no association with MASLD incidence. Among the dysglycemic subjects, there was no observed association between VAI and the incidence of MASLD or the progression of fibrosis. In conclusion, the BMI, WC, and the VAI are associated with an increased risk of progression to moderate-to-advanced liver fibrosis in the general population. However, the VAI does not perform better than the BMI and WC measurement.

## 1. Introduction

Nonalcoholic fatty liver disease (NAFLD) currently represents the most prevalent cause of chronic liver disease in the western world [1,2,3]. Recently, its nomenclature has been redefined by an international panel of experts in this pathology with the term MASLD (metabolic dysfunction-associated steatotic liver disease) [4]. In this new nomenclature, metabolic dysfunction takes on greater emphasis as a factor associated with its etiopathogenesis. MASLD is a general term that includes a wide spectrum of lesions ranging from benign hepatic steatosis or simple steatosis to nonalcoholic steatohepatitis (or currently MASH: metabolic dysfunction-associated steatohepatitis), liver fibrosis, and, in some cases, an evolution into liver cirrhosis and hepatocellular carcinomas [5]. This entity is considered a part of a systemic disease, and is strongly associated with obesity, type 2 diabetes mellitus (T2D), insulin resistance (IR), arterial hypertension and atherogenic dyslipidemia [6,7,8].

In relation to body adiposity and liver fat, it has been discovered that there is a direct relationship between visceral adiposity, IR, and hepatic fat content [9]. Moreover, visceral adiposity is associated with more advanced stages of fatty liver disease such as inflammation and fibrosis [10]. Indeed, our group recently discovered that the presence of abdominal obesity and dysglycemia were the main metabolic risk factors associated with moderate-to-advanced liver fibrosis development over time in the general population [11]. The abdominal obesity indices used in routine clinical practice include the body mass index (BMI) and waist circumference (WC) measurement. It is well known that BMI alone is not sufficient to accurately assess or manage the cardiometabolic risk associated with increased adiposity in adults. The limitation of the BMI in comprehensively assessing cardiometabolic risk is, in part, due to its inadequacy as an isolation biomarker for overall body composition, particularly in accurately measuring central abdominal fat mass [12]. On the other hand, WC is used as a marker of visceral adiposity and, despite not discriminating between fat in the visceral and subcutaneous compartments, it reflects visceral adiposity better than the BMI [13]. Growing evidence suggests that visceral adiposity tissue (VAT) is a key conduit through which obesity predicts health risk. In this sense, a novel marker of visceral adiposity content and dysfunction has been proposed: the visceral adiposity index (VAI). This index has been shown to be a more accurate surrogate marker than the classical anthropometric indices in predicting metabolic disorders linked to IR [14]. Epidemiological studies suggest a positive association between the VAI and the prevalence of MASLD [15,16,17]. However, there are controversial data regarding the association between the VAI and liver fibrosis [16,18,19].

To our knowledge, there is no information about the relationship between abdominal obesity indexes, including the VAI, and the risk of liver stiffness progression in the general population. The aim of the present study was to investigate the association between abdominal obesity indexes (the BMI, WC, and the VAI) with the progression to moderate-to-advanced liver fibrosis assessed by transient elastography (TE) in a well-characterized asymptomatic general population. In addition, we analyzed the relationship between these abdominal obesity indexes with the prevalence of MASLD and liver fibrosis.

## 2. Materials and Methods

### 2.1. Study Design and Participants

This large-scale community-based study encompasses both cross-sectional and longitudinal follow-up data, based on a previous descriptive population study about liver fibrosis prevalence [20] and fibrosis progression [11]. Research was carried out in various communities located in the northern part of the Barcelona metropolitan area. In the cross-sectional study, participants were recruited between April 2012 and January 2016. For the longitudinal follow-up, a subsequent cross-sectional assessment was conducted 4 years later, between October 2016 and December 2019.

Participants in the cross-sectional study were randomly identified from 162,950 subjects aged 18–75 years from the registries of the primary health care centers of the municipalities included in the study. Randomly selected individuals were contacted by phone and invited to take part in the study. Patients with a current history of liver disease, including cholestasis, hepatitis C or B virus infection, and high-risk alcohol consumption were excluded from the study. High-risk alcohol consumption was defined as >21 standard beverage units (SBUs) per week in men and >14 SBUs in women [21]. Additional exclusion criteria encompassed active malignancy, severe comorbidities (congestive heart failure > 2 via the New York Heart Association’s standards, chronic obstructive pulmonary disease defined as a Global Initiative for Chronic Obstructive Lung Disease score > 2, chronic kidney disease requiring dialysis, previous organ transplantation, and severe neurological diseases) or admission in long-term care facilities.

After securing informed consent, the subsequent procedures were initiated. First, a detailed medical history was taken, including alcohol consumption, anthropometric measurements, body weight, height, BMI, WC, and arterial pressure. Secondly, biological measures were conducted following a 12 h fast, which included assessments of the liver biochemistry, hepatitis B and C virus markers (HBsAg and anti-HCV), serum fasting glycemia, glycosylated hemoglobin (HbA1c), serum creatinine, serum ferritin, and the serum lipid profile (total cholesterol (TC), LDL cholesterol (LDL-C), HDL cholesterol (HDL-C), and triglycerides (TGs). Additionally, the concentration of remaining cholesterol (expressed in mg/dL) was calculated using the following equation: total cholesterol—LCL-C—HDL-C. Lastly, serological markers for the diagnosis of steatosis (the fatty liver index—the FLI) and fibrosis (the NAFLD fibrosis score (NFS) and Fibrosis-4 index (FIB-4)) were determined. An elastography with a liver stiffness measurement (LSM) was performed at baseline and at the end of the follow-up period.

### 2.2. Definitions

The presence of NAFLD in this well-established cohort was determined based on standard diagnostic criteria, including one or more positive findings on the fatty liver index (FLI ≥ 60), abdominal ultrasound, or liver biopsy [11,20,22,23]. Nevertheless, considering the new nomenclature, we opted to adopt the term MASLD given the indications that 98% of the current registry cohort of NAFLD patients would align with the new MASLD criteria [4].

The presence of liver fibrosis was determined via TE. This was performed via the use of the Fibroscan^®^ system (402, Echosens^®^, Paris, France) by three trained specialist liver nurses. The XL probe was unavailable, so we only used the M probe. Moderate-to-advanced liver fibrosis was defined by LSM values ≥ 8.0 kPa according to other epidemiological studies [24,25]. We defined fibrosis progression as the change from one stage to another (LSM < 8 to ≥8 kPa) but requiring, at least, an increase of 1 kPa between the LSM measurement at baseline and follow-up, based on previous studies [26,27].

NFS and FIB-4 scores were computed, with a high risk of significant fibrosis determined based on the cut-off points described in the original publications: FIB-4 > 2.67 and NFS ≥ 0.676, respectively [28,29].

Obesity was determined by a BMI ≥ 30 kg/m^2^ and being overweight was determined to be in the ≥25 kg/m^2^ to 29.9 kg/m^2^ BMI range. Weight circumference (cm) was obtained by measuring the abdominal circumference in the intermediate area between the last costal arch and the iliac crest with a tape measure, measured in a horizontal plane with the abdomen relaxed. Abdominal obesity was determined by values > 102 cm in men and >88 cm in women. The value of the VAI was calculated as the following formula based on clinical parameters, and was sex-specific [14]: Males: [WC (cm)/(39.68 + 1.88 × BMI (kg/m^2^))] × [TG (mmol/L)/1.03] × [1.31/HDL-C (mmol/L)]; Females: [WC (cm)/(39.58 + 1.89 × BMI (kg/m^2^))] × [TG (mmol/L)/0.81] × [1.52/HDL-C (mmol/L)]. The different VAI quartiles were calculated by sex, using the data from our sample.

The diagnosis of type 2 diabetes mellitus (T2D) and prediabetes was established either through a registered diagnosis in clinical records or by meeting specific criteria: levels of HbA1c ≥ 6.5% or fasting glucose ≥ 126 mg/dL and Hb1Ac levels of 5.7–6.4% or a fasting glucose of 100–125 mg/dL, respectively. For analysis purposes, we categorized both subjects with T2D and prediabetes under the predefined term of dysglycemia.

### 2.3. Statistical Analysis

Descriptive analysis included frequencies and percentages for categorical variables, means and standard deviation for continuous variables. The comparison of baseline vs. follow-up means was assessed via a paired Student’s *t*-test and a comparison of percentages via McNemar’ test. The linear trend between the different obesity index categories (the VAI and BMI) and the prevalence or incidence of MASLD (FLI ≥ 60) or liver fibrosis (TE ≥ 8 kPa) was assessed using the Cochran–Armitage test for trends. When the prevalences were compared between two categories (for abdominal obesity or comparing overweight to obese) chi-squared tests were used.

The relationship between the baseline VAI (4th quartile vs. the 1st, 2nd, and 3rd quartiles), central obesity (BMI ≥ 30 kg/m^2^), and abdominal obesity, and the prevalence of MASLD or hepatic fibrosis was assessed using logistic multivariate models. In these models, baseline MASLD or hepatic fibrosis served as the dependent variables, while the VAI, central obesity, and abdominal obesity were the main explanatory variables (each assessed separately, not in the same model). The analysis was adjusted for sex, age, and dysglycemia. Additionally, a sensitivity analysis was conducted specifically in individuals with baseline dysglycemia.

In the incidence of MASLD (from a baseline FLI < 60 to a follow-up FLI ≥ 60) or fibrosis (from a baseline TE < 8 kPa to a follow-up TE ≥ 8 kPa with an at least 1 kPa increase), subjects with a baseline FLI ≥ 60 or a TE ≥ 8 kPa were excluded, respectively. As the date of the incident of MASLD or fibrosis is the date when the follow-up study was performed (the second cross-sectional study), multivariate logistic regression was used to assess the relationship of new MASLD or fibrosis diagnosis (dependent variables) and the VAI, obesity and abdominal obesity (main explanatory variables), adjusted by sex, baseline age, and dysglycemia. A sensibility analysis was performed, in addition, using baseline dysglycemia and/or BMI. Interaction models of the three explanatory variables (the VAI, BMI, and abdominal obesity) were performed using likelihood ratio tests.

The comparison of the change in liver stiffness measurements between the baseline and the follow-up by the groups (using either one of the obesity variables or their combination) was assessed via a *t*-test (binary variables) and an ANOVA (>2 categories). The comparison of fibrosis or MASLD incidence by groups was assessed using chi-squared tests, excluding those with basal fibrosis or MASLD, respectively.

The comparison of the changes in the BMI between baseline and follow-up (a decrease if the BMI was reduced by ≥1 kg/m^2^, an increase if the BMI was incremented by ≥1 kg/m^2^, of stable if otherwise) and the changes in MASLD or fibrosis status was tested using chi-squared tests.

All comparisons were bilateral, and the significance was 0.05. Statistical analysis was performed using the Stata v18 statistical package.

## 3. Results

### 3.1. Characteristics of the Study Population

Of the 4866 subjects invited, 3076 agreed to participate in the cross-sectional study (a participation rate of 63.2%). After excluding subjects who did not meet the inclusion criteria, the final number of subjects 2840. Of these, 1363 were not included in the longitudinal follow-up for different reasons, as shown in the flow diagram in Figure 1.

Therefore, the population included in this study was 1403 subjects (95% of whom were Caucasian) in whom two LSMs were obtained, the first at baseline and the second one at the end of the follow-up (a median 4.2 years later; a range of 3.0–5.5). The clinical and biochemical characteristics of the whole cohort at baseline and follow-up are described in Table 1.

At baseline, the prevalence of MASLD and liver fibrosis by TE (LSM ≥ 8.0 kPa) was 36% and 5%, respectively. Changes in longitudinal LSMs according to the stage of the baseline TE (<8.0 kPa or ≥8.0) are represented in Appendix A. At follow-up, 3% of subjects had a change in LSMs from <8.0 kPa to ≥8.0 kPa. Additionally, we found that 14% of subjects who met the MASLD criteria at baseline exhibited a regression at follow-up.

### 3.2. Abdominal Obesity Indices: Their Relationship with MASLD and Liver Fibrosis Prevalence

The prevalences of MASLD and moderate-to-advanced liver fibrosis (LSM ≥ 8.0 kPa) according to the BMI categories, WC measurements, and the VAI quartiles are represented in Figure 2. The highest prevalence of MASLD and moderate-to-advanced fibrosis was observed among subjects with obesity (83% for MASLD and 12% for fibrosis), with a VAI Q4 (67% for MASLD and 11% for fibrosis), and among those with abdominal obesity (59% for MASLD and 8.2% for fibrosis). In all scenarios, MASLD and fibrosis prevalences increased as the different obesity indexes went increased (*p* < 0.001).

The prevalence of MASLD and moderate-to-advanced liver fibrosis, stratified by VAI quartiles in relation to the BMI categories is shown in Figure 3. We observed a linear increase in the prevalence of MASLD as the VAI quartile increases, both in subjects with obesity and who were overweight. Nevertheless, concerning the prevalence of moderate-to-advanced liver fibrosis, no clear linear increasing trend was observed among patients who were overweight. Instead, a higher percentage of liver fibrosis was observed across the VAI quartiles, particularly in those within highest quartile, in subjects with obesity compared to those who were overweight.

In the multivariate analysis adjusted by dysglycemia, sex, and age, a VAI of Q4 and a BMI ≥ 30 kg/m^2^ were significantly associated with the presence of MASLD (OR 5.1 [CI] 3.9–6.8 and OR 33.9 [CI] 24–48) and moderate-to-advanced liver fibrosis (OR 3.5 [CI] 2.1–5.9 and OR 8.8 [CI] 4.7–17). For abdominal obesity, results were similar to those of the BMI (Table 2). No interaction effects were found between the VAI, BMI, and abdominal obesity.

### 3.3. Abdominal Obesity Indices: Their Relationship with MASLD Incidence and Liver Fibrosis Progression

The progression of liver stiffness and the incidence of MASLD based on the BMI categories, the VAI quartiles, and the WC measurement for AO are depicted in Figure 4. A high incidence of MASLD was noted among subjects with a BMI ≥30 kg/m^2^ or abdominal obesity. Conversely, those in the higher VAI quartile, whether they had obesity or were displaying abdominal obesity, exhibited a greater progression to moderate-to-advanced liver fibrosis.

Changes in MASLD incidence were significantly associated with changes in BMI (*p* < 0.001). Specifically, the incidence of MASLD was higher in individuals who experienced a BMI increase of at least 1 kg/m^2^, while a regression from MASLD to non-MASLD was more common in those who had a BMI decrease of at least 1 kg/m^2^. A similar trend was found in the relationship between the BMI and liver fibrosis changes (*p* = 0.001) (Appendix A).

Combining the BMI–VAI indices revealed a higher percentage of progression to moderate-advanced liver fibrosis in subjects with obesity, particularly in those with the highest VAI quartile (9.3% for Q4 vs. 5.3% for Q1–Q3). Moreover, when all three highest categories of abdominal indices (a BMI ≥ 30 kg/m^2^, a VAI of Q4, and abdominal obesity) were analyzed collectively, the progression to moderate-to-advanced fibrosis exhibited a greater increase than when each of them was considered individually (Table 3).

Following adjustments for dysglycemia, age, and sex, the incidence of MASLD showed significant associations with a BMI ≥ 30 kg/m^2^ and WC measurements (OR 2.8 [CI] 1.7–4.6 and OR 3.1 [CI] 2.2–4.4). Nevertheless, no association was observed with the highest quartile of the VAI (OR 1.2 [CI] 0.8–1.8). In relation to liver fibrosis, the presence of a VAI of Q4, a BMI ≥ 30 kg/m^2^, and abdominal obesity were positively associated with fibrosis progression (OR 2.0 [CI] 1.0–4.1; OR 4.7 [CI] 2.3–9.4 and OR 7.9 [CI] 3.2–20) (Table 2). We did not observe any significant interaction between the three indices and MASLD incidence or fibrosis progression.

### 3.4. Abdominal Obesity Indices in Dysglycemic Subjects: Their Relationship with MASLD and Liver Fibrosis Prevalence

At baseline, 357 subjects had dysglycemia (170 with T2D and 187 with prediabetes). Of these subjects, 186 did not have obesity (BMI < 30 kg/m^2^). In the multivariate analysis, a BMI ≥ 30 kg/m^2^, abdominal obesity, and a VAI of Q4 were significantly associated with the presence of MASLD (OR 32 [CI] 16–64; OR 42 [CI] 17–109; OR 4.0 [CI] 2.5–6.4) and moderate-to-advanced liver fibrosis (OR 7.3 [CI] 3.1–17; OR 8.4 [CI] 2.8–25; OR 7.1 [CI] 3.2–16) in subjects with dysglycemia, respectively (Table 2).

In subjects with dysglycemia but without obesity, multivariate analysis revealed a significant association between the presence of a VAI of Q4 and the prevalence of fibrosis (OR 12 [CI] 1.4–110). Although the odds ratio was higher compared to the overall cohort and to those with dysglycemia, the confidence intervals overlapped with those observed in those groups.

### 3.5. Abdominal Obesity Indices in Dysglycemic Subjects: Their Relationship with MASLD Incidence and Liver Fibrosis Progression

The BMI and WC were significantly associated with the presence of new cases of MASLD and moderate-to-advanced liver fibrosis (OR 6.7 [CI] 1.7–26; OR 6.3 [CI] 2.0–20 and OR 3.2 [CI] 1.3–7.9; OR 10 [CI] 2.3–46, respectively), while no association was observed with the VAI of Q4 (OR 1.0 [CI] 0.5–2.3 and OR 1.8 [CI] 0.7–4.5) (Table 2). Furthermore, upon analyzing subjects with dysglycemia but without obesity, the VAI was not found to be associated with the occurrence of new cases of MASLD or with fibrosis progression (OR 1.3 [CI] 0.6–3.0 and OR 2.8 [CI] 0.4–22).

## 4. Discussion

To our knowledge, this study is the first to investigate the association between abdominal obesity indices, with a particular emphasis on the VAI, and the progression of liver fibrosis assessed via transient elastography in the general population. This extensive community-based study incorporates both cross-sectional and longitudinal data. In the longitudinal analysis, our findings reveal a noteworthy association between various abdominal indices utilized in routine clinical practice—namely, the BMI, WC, and the VAI—and the progression of liver fibrosis. Furthermore, abdominal indices emerge as independent risk factors associated with the prevalence of both MASLD and liver fibrosis. Data obtained from population-based studies indicate a high prevalence of significant liver fibrosis, primarily linked to the presence of MASLD [20]. The prevalence of MASLD is widely recognized to be strongly linked to the rise in obesity and T2D [30]. Several studies have highlighted the critical role of body fat distribution in determining long-term outcomes and mortality, with evidence supporting a correlation between visceral adiposity and both liver-fat accumulation and the progression of liver disease in MASLD [10,31,32,33]. Magnetic resonance evaluation demonstrates a direct relationship of visceral adipose tissue with the severity of inflammation and liver fibrosis in individuals with MASLD [34]. Additionally, a novel score, known as the VAI, has emerged as a surrogate marker for visceral fat distribution and dysfunction [35]. Unlike the BMI and WC, the VAI is proposed as an index capable of assessing liver dysfunction and insulin sensitivity [14].

Data from our cross-sectional study reveal a steady increase in MASLD and liver fibrosis prevalence with higher quartiles of the VAI and the BMI, coupled with the presence of abdominal obesity. Thus, in our overall cohort, subjects with obesity and the highest quartile of the VAI exhibited a greater prevalence of MASLD and liver fibrosis. The VAI score, by incorporating both clinical and metabolic parameters, aims to capture visceral fat dysfunction, including phenomena such as the release of adipocytokines, heightened lipolysis, and elevated plasma free fatty acids. These aspects are not fully represented when considering WC measurements, the BMI, TGs, and HDL-C individually [19]. Additionally, our data also reveal a significant association between the highest quartile of the VAI and the prevalence of moderate-to-advanced liver fibrosis. These findings remained statistically significant among subjects with dysglycemia, including those without obesity. In a previous publication, we reported that factors independently associated with increased liver stiffness measurements in the general population included abdominal obesity, HDL-C, and TG levels, which are parameters incorporated into the VAI formula [20]. Overall, our findings indicate that the VAI is an independent risk factor for MASLD, consistent with other studies [18,35,36], and for moderate-to-advanced liver fibrosis. However, despite the VAI incorporating both the BMI and WC into its formula, it was not more potent than these two classic anthropometric measures separately.

There is limited research published on the association between the VAI and the presence of fibrosis, primarily comprising cross-sectional studies conducted in subjects diagnosed with MASLD, with liver biopsy being the predominant diagnostic method utilized. Data from previous studies indicate a strong association between different indices of obesity, including the VAI, and the presence of MASLD/MASH, although its relationship with liver fibrosis remains controversial [10,15,16,17,18,19,36,37,38]. Several studies conducted of subjects with chronic liver disease due to virus C found a correlation between the VAI and the severity of steatosis and necroinflammatory activity [39,40]. In line with our findings, Petta et al. reported that in subjects with MASLD hospitalized in a liver unit, the VAI was not only independently correlated with significant fibrosis, as determined via liver biopsy, but also that the prevalence of significant fibrosis rose with increasing VAI values. In contrast, in the study conducted by Vongsuvanh et al. [19], the VAI was not associated with steatosis, inflammation, or fibrosis in subjects with biopsy-proven MASLD. In this same study, WC was also significantly correlated with liver fibrosis, whereas TGs and HDL-C were not of any predictive value. These authors speculated that the direct pro-inflammatory effects of visceral fat might play a more significant role than dyslipidemia. In addition, in a recent population-based study of subjects with MASLD from the US, the VAI was positively associated with the prevalence of MASLD, but not with liver fibrosis [35]. We hypothesized that the discrepant findings of these studies may be due to differences in the metabolic conditions of the subjects involved. Additionally, we propose that these conflicting results could underscore an inherent limitation of the VAI formula. The VAI score places considerable emphasis on TG and HDL-C levels, both serving as multiplication factors in the equation. However, this effect may not manifest in cohorts where TG and HDL-C levels are not substantially different from normal levels. Notably, in our population study, approximately 10% of participants exhibited atherogenic dyslipidemia. Moreover, as suggested by other authors, TGs and HDL-C are strongly linked to cardiovascular risk and may not fully capture the pathophysiological changes of the inflammatory process leading from steatosis to steatohepatitis and fibrosis. Additionally, there may be variables within the inflammatory process that are essential for fibrosis progression and are not included in the VAI [19,38,41].

Further, we analyzed the link between abdominal obesity indices, with particular emphasis on the VAI index, and the progression to moderate-to-advanced liver fibrosis. It is known that progressive liver fibrosis is the main determinant of long-term outcomes in MASLD [30]. Identifying individuals at a heightened risk of liver fibrosis progression is particularly crucial, as it can lead to significant benefits in terms of reducing liver-related morbidity and mortality. Previous population studies have shown that both abdominal obesity measured through WC and dysglycemia are risk factors for fibrosis progression [11]. Moreover, in subjects with MASLD, obesity has been associated with an increased risk of fibrosis progression over time [27].

One of the main findings of the present study is the identification of an independent association between higher quartiles of the VAI, along with the BMI and WC measurements, and the progression to moderate-to-advanced liver fibrosis during the follow-up. However, like what occurred in the cross-sectional study, the VAI was no more useful for detecting fibrosis progression compared to the BMI and WC measurements. Nevertheless, it is worth noting that when examining all three highest categories of abdominal indices together, the progression to moderate-to-advanced fibrosis showed a more pronounced increase compared to when each was evaluated individually. Therefore, combining the three indices of abdominal obesity in clinical practice may contribute to better identifying those patients who are at risk of progressing to significant liver fibrosis. Regarding MASLD, we found an association with a BMI ≥ 30 kg/m^2^ and WC measurements, but not with the highest quartiles of the VAI. As far as we know, there is only one large-scale study conducted by Xen C et al. [36] in North China, that investigates the association between abdominal obesity indices, including the VAI, and the risk of developing MASLD over time. Nonetheless, the progression of liver fibrosis was not evaluated in this study. In the longitudinal population study, a newly developed VAI specific to Chinese adults (CVAI) demonstrated the strongest associations with the incidence of MASLD among the abdominal obesity indices. However, like our findings, when the classical VAI score was examined, it was not associated with the risk of developing MASLD.

It is known that dysglycemia is a risk factor for MASLD and liver fibrosis progression [11]. In the cross-sectional study, we observed a strong association between abdominal indices, including the VAI, and the presence of MASLD as well as moderate-to-advanced liver fibrosis in subjects with dysglycemia, mirroring the trends observed across the entire cohort. Notably, this association was also evident in dysglycemic subjects who were not classified as having obesity. Findings from a case–control study conducted within a Chinese population with MASLD revealed that lean individuals with MASLD exhibited lower BMI and WC measurements compared to control subject who were considered overweight or obese. However, their VAI levels were notably higher [42]. It is noteworthy that MASLD is not exclusive to individuals with excess weight; it also occurs in lean individuals [43,44], and a considerable proportion of them exhibited IR [45].

However, data from our study reveal that the BMI and WC measurements were the only risk factors associated with liver fibrosis progression during follow-up. Surprisingly, we did not identify any association between being in the highest quartile of the VAI and the progression to liver fibrosis in subjects with dysglycemia. There might be several reasons for this finding. First, it is important to note that the study was carried out within a community setting, resulting in having a relatively low percentage of subjects with dysglycemia (with 52% of them having pre-diabetes). Furthermore, the majority exhibited optimal glycemic control, as evidenced by a median HbA1c of 6.4% at baseline, which remained at follow-up. Second, the percentage of the progression to moderate-to-advanced liver fibrosis was relatively low during the follow-up period in the dysglycemic group (changes in LSMs from <8 kPa to ≥8 kPa: 8%). It is possible that a longer duration of follow-up is necessary to detect significant changes in LSMs. Third, although data about treatments with statins of fibrates were not available, it is expected that subjects with dysglycemia take more lipid-lowering agents affecting their levels of TGs and HDL-C, both of which are key parameters in the VAI formula. Notably, our recent study revealed that atherogenic dyslipidemia independently correlated with the progression to moderate-to-advanced liver fibrosis in the general population; however, this association was not observed within the dysglycemic subgroup [11]. These results might suggest the presence of additional mechanisms that directly link visceral fat to MASLD, independent of insulin resistance or steatosis, and contribute to the progression of liver fibrosis, mechanisms that are not accounted for in the VAI formula [37].

The main strength of our study lies in the large and meticulously characterized general population cohort from which we derived our results. However, our study is subject to certain limitations, which we outline below:(1)We did not compare our results with histology data to evaluate fibrosis progression, which is reasonable given the challenges associated with such testing, particularly in a study involving the general population. Instead, we utilized TE, an invasive yet straightforward and reproducible test, to assess the presence and progression of fibrosis.(2)The observed changes in LSM values during follow-up were relatively minor. This could be attributed to several factors. Firstly, our study does not target specific hospital, clinic, or specialized unit populations where individuals with known liver pathologies are specifically referred. Secondly, we excluded individuals with a history of liver disease, including those with viral hepatitis or hazardous alcohol consumption. Thirdly, a longer follow-up period may be necessary to detect significant changes in LSMs.(3)The XL probe was unavailable for use in our study. Although this probe might have potentially reduced the failure rate, particularly among the subjects with obesity, we believe it would not have significantly affected our main findings. This is because we excluded patients with obesity with invalid TE readings or those in whom the technique could not be performed, resulting in a low rate of unreliable liver stiffness measurements (only 1.5%).(4)Data on the medication used by the study participants, such as glucose and lipid-lowering drugs, was not available, which could influence the metabolic characteristics of certain patient subgroups, particularly those with dysglycemia.(5)The HOMA was not utilized in our cohort to assess insulin resistance, which could have provided a more precise evaluation of the relationship between the VAI and insulin resistance in MASLD. Similarly, we lack data on inflammatory parameters and adipokines, which could have helped establish a connection between the VAI and the development of liver fibrosis.(6)Although we had sufficient statistical power for the primary objectives of our study, the sample size might be inadequate for specific subgroup analyses, posing a potential limitation.

## 5. Conclusions

We present, for the first time, evidence that traditional abdominal indices such as the BMI and WC measurements, along with the novel VAI index, are associated with an increased risk of progression to moderate-to-advanced liver fibrosis in the general population. However, in our cohort, the VAI was not more powerful than the BMI and WC measurements in predicting fibrosis progression. Although, the simplicity of measuring the WC and BMI, along with the assessment of TGs and HDL-C, renders the VAI a readily applicable index for evaluating visceral fat dysfunction. Thus, the VAI could prove to be a valuable tool in daily clinical practice and population studies for assessing the cardiometabolic risk associated with visceral obesity. Additional research is required to validate the significance of obesity indices in the advancement of liver fibrosis and their applicability in standard clinical settings.

## Figures and Tables

**Figure 1 nutrients-16-01586-f001:**
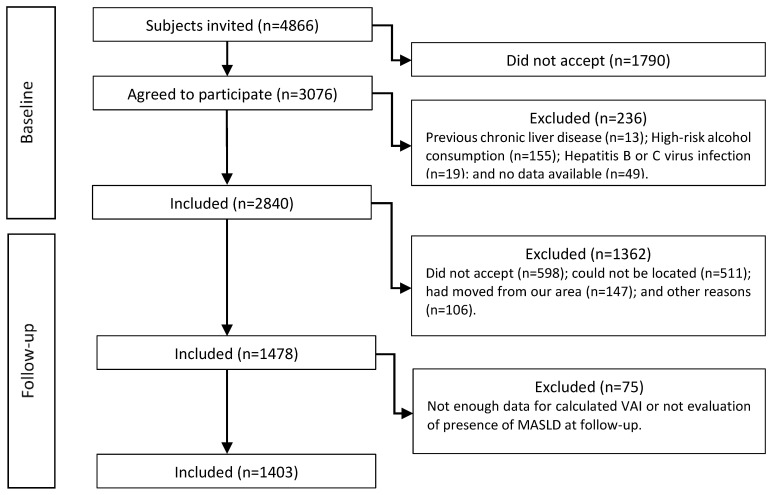
The study’s flow chart.

**Figure 2 nutrients-16-01586-f002:**
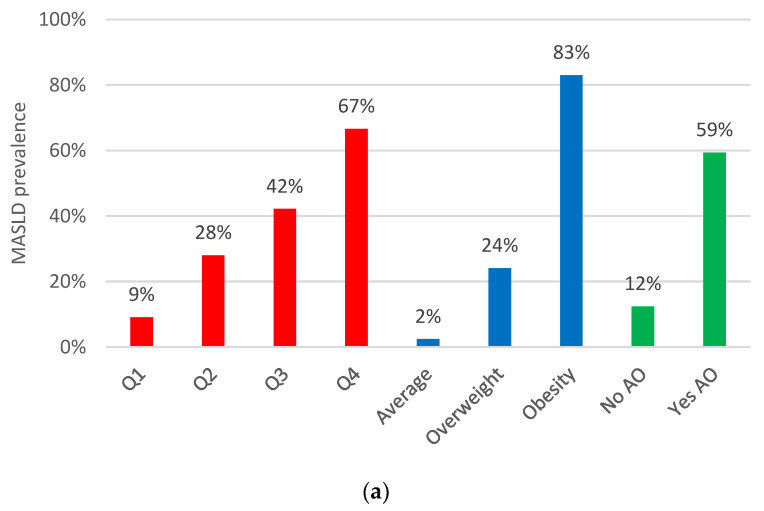
The prevalence of MASLD (**a**) and moderate-to-advanced liver fibrosis (LSM ≥ 8 kPa) (**b**) according to different abdominal indices. The VAI is represented by red bars, the BMI by blue bars, and abdominal obesity by green bars. Overweight: IMC < 30 kg/m^2^, and obesity: IMC > 30 kg/m^2^. Abbreviations: VAI (visceral adiposity index); BMI (body mass index); AO (abdominal obesity). Either for MASLD of fibrosis prevalences, the Cochran–Armitage test for trend *p*-values were <0.001 for the VAI and BMI, and the chi-squared *p*-values were 0.001 for abdominal obesity.

**Figure 3 nutrients-16-01586-f003:**
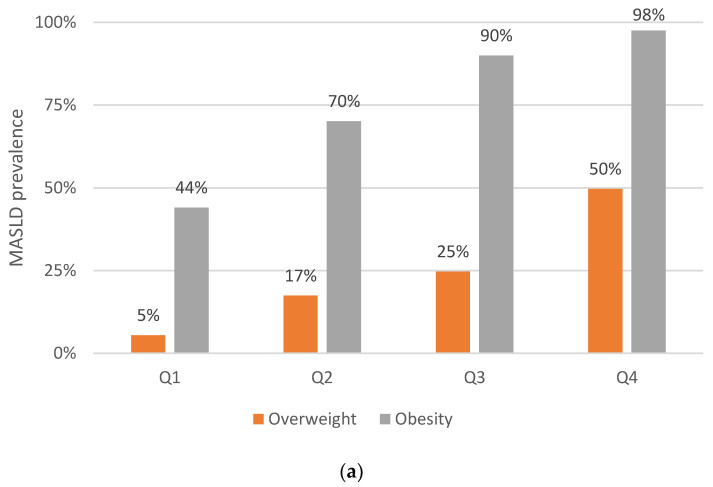
The prevalence of MASLD (**a**) and moderate-to-advanced liver fibrosis stratified (**b**) by the VAI quartiles in relation to the BMI categories. Orange bars represented subjects who were overweight (IMC < 30 kg/m^2^) and grey bars represented subjects with obesity (IMC > 30 kg/m^2^). Abbreviations: VAI (visceral adiposity index); BMI (body mass index); AO (abdominal obesity). For MASLD prevalence, the Cochran–Armitage test for trend *p*-values were <0.001 across the VAI quartiles, both for patients who were overweight and patients with obesity. Regardless of the quartile of the VAI the subjects belonged to, the prevalence of MASLD was consistently found to be significantly lower than 0.001 among both patients who were overweight and patients who were obese. For fibrosis prevalence, the Cochran–Armitage test for trend *p*-value was 0.076 across the VAI quartiles for patients who were overweight, and <0.001 for patients with obesity. Comparing the fibrosis prevalence between the patients who were overweight and those who were obese, *p* was 0.004 (among Q1), 0.137 (Q2), 0.001 (Q3), and <0.001 (Q4).

**Figure 4 nutrients-16-01586-f004:**
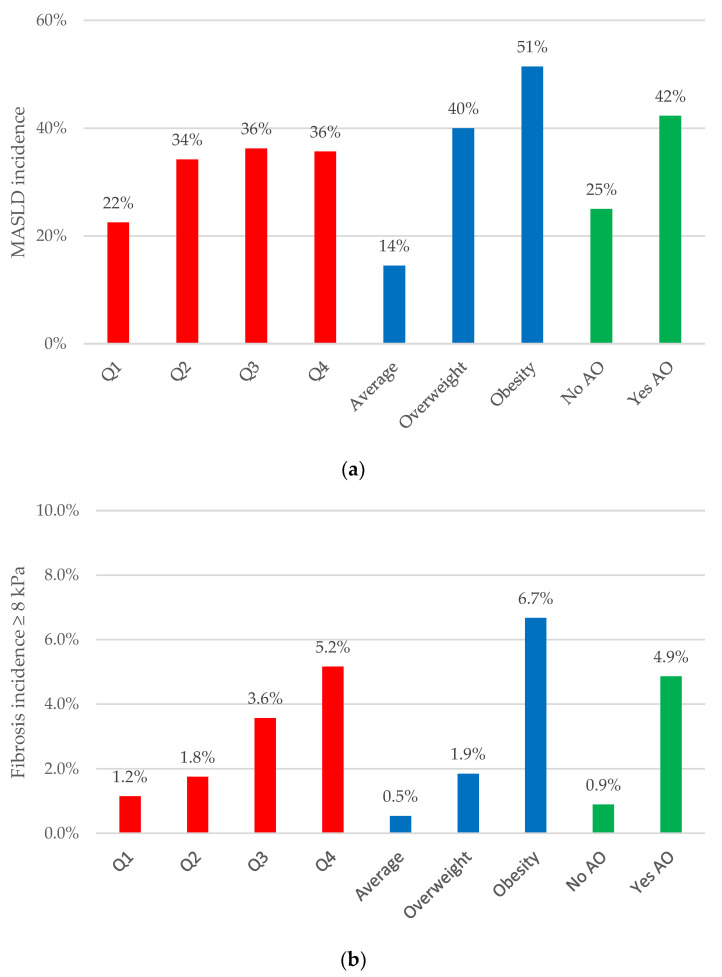
The incidence of MASLD and liver fibrosis progression (LSM < 8 to ≥8 kPa) according to different abdominal indices. (**a**) New cases of MASLD across the quartiles of the VAI, the BMI categories, and and AO determined by WC. (**b**) Liver fibrosis progression across the quartiles of the VAI, the BMI categories, and AO determined by WC. To be considered a progression, a minimum increase of 1 kPa was required. Overweight: IMC < 30 kg/m^2^. Obesity: IMC > 30 kg/m^2^. Abbreviations: VAI (visceral adiposity index); BMI (body mass index); AO (abdominal obesity). For MASLD or fibrosis incidences, the Cochran–Armitage test for trend *p*-values were <0.001 for the VAI and the BMI, and the chi-squared *p*-values were 0.001 for abdominal obesity.

**Table 1 nutrients-16-01586-t001:** Characteristics of the study population at baseline and follow-up.

	Total (*n* = 1403)			
	Baseline		Follow-Up	
	*n*/Mean	sd/%	*n*/Mean	sd/%
Age, years	56	±11	60	±11
Female, *n* (%)	873	(62%)	873	(62%)
VAI	1.7	1.4	1.7	1.1
Body mass index, kg/m^2^	28	±5	28	±5
Abdominal obesity, *n* (%)	718	(51%)	767	(55%)
T2D, *n* (%)	170	(12%)	231	(16%)
Prediabetes, *n* (%)	187	(13%)	149	(11%)
Dysglycemia *, *n* (%)	357	(25%)	380	(27%)
Glucose, mg/dL	99	±24	101	±23
Glycated hemoglobin (%)	5.7	0.7	5.7	0.7
Triglyceride, mg/dL	120	±72	115	±57
Total cholesterol, mg/dL	214	±38	208	±39
LDL-cholesterol, mg/dL	135	±33	130	±34
HDL-cholesterol, mg/dL	56	±13	55	±13
Cholesterol remnants †, mg/dL	23	±13	23	±14
Atherogenic dyslipemia §, *n* (%)	132	(9%)	136	(10%)
ALT and/or AST > 40 U/L, *n* (%)	108	(8%)	100	(7%)
FLI ¶	47	±28	49	±28
FLI ≥ 60, *n* (%)	490	(35%)	482	(37%)
MASLD, *n* (%)	511	(36%)	677	(51%)
Mean liver fibrosis by LSM (kPa)	4.9	2.2	4.9	2.1
Liver fibrosis by Fibroscan ≥ 8.0 kPa, *n* (%)	68	(5%)	62	(4%)
Liver fibrosis by Fibroscan ≥ 9.2 kPa, *n* (%)	35	(2%)	40	(3%)
FIB-4 > 2.67, *n* (%)	24	(2%)	53	(4%)
High NFS, *n* (%)	16	(1%)	73	(6%)

Data are *n* (%) or mean ± SD. Abbreviations: VAI: visceral adiposity index; T2D: type 2 diabetes mellitus; ALT, alanine aminotransferase; AST, aspartate aminotransferase; FLI, fatty liver index; HDL, high-density lipoprotein; LDL, low-density lipoprotein; LSM: liver stiffness measurement (by elastography); MASLD, non-alcoholic fatty liver disease. FIB-4: Fibrosis-4 index. NFS: NAFLD fibrosis score. * Dysglycemia: T2D and prediabetes. † Cholesterol remnants: total cholesterol- LDL-C-HDL-C. § Atherogenic dyslipidemia is defined by triglycerides > 150 mg/dL and HDL-C < 40 mg/dL in men and <50 mg/dL in women. ¶ FLI (fatty liver index) estimates the amount of fat in the liver and includes the body mass index, waist circumference, and the amounts of serum gamma-glutamyltransferase and triglycerides.

**Table 2 nutrients-16-01586-t002:** The multivariate analysis of the abdominal obesity indices associated with MASLD and moderate-to-advanced liver fibrosis in the cross-sectional and longitudinal study in the global cohort and in subjects with dysglycemia.

	Cross-Sectional				Longitudinal			
	Overall		Dysglycemia		Overall		Dysglycemia	
	MASLD	Fibrosis	MASLD	Fibrosis	MASLD	Fibrosis	MASLD	Fibrosis
BMI	33.9 (24–48) *p* < 0.001	8.8 (4.7–16) *p* < 0.001	32 (16–64) *p* < 0.001	7.3 (3.1–17) *p* < 0.001	2.8 (1.7–4.6) *p* < 0.001	4.7 (2.3–9.4) *p* < 0.001	6.7 (1.7–26) *p* < 0.006	6.3 (2.0–20) *p* < 0.002
VAI Q4	5.1 (3.9–6.8) *p* < 0.001	3.5 (2.1–5.9) *p* < 0.001	4.0 (2.5–6.4) *p* < 0.001	7.1 (3.2–16) *p* < 0.001	1.2 (0.8–1.8) *p* = 0.454	2.0 (1.0–4.1) *p* = 0.041	1.0 (0.5–2.3) *p* = 0.905	1.8 (0.7–45) *p* = 0.231
Abdominal obesity	37.5 (23–60) *p* < 0.001	7.6 (3.6–16) *p* < 0.001	42 (17–109) *p* < 0.001	8.4 (2.8–25) *p* < 0.001	3.1 (2.2–4.4) *p* < 0.001	7.9 (3.2–20) *p* < 0.001	3.2 (1.3–7.9) *p* = 0.012	10 (2.3–46) *p* = 0.003

At baseline, adjusted by dysglycemia, sex, and age. Values had a CI of 95%. Abbreviations: VAI (visceral adiposity index); BMI (body mass index).

**Table 3 nutrients-16-01586-t003:** The association between various baseline abdominal indices, either independently or in combination with the incidence of MASLD and the progression of fibrosis.

Index	*n*	∆ LSM (kPa)	F < 8 to F ≥ 8 (%)	MASLD Incidence (%)
**VAI**				
Q1–Q3	1053	0.14	2.2	29.9
Q4	350	−0.13	5.2	35.7
**BMI**				
Non-obese	973	0.13	1.4	28.8
Obese	430	−0.04	6.7	51.4
**Abdominal obesity**				
No	685	0.10	0.9	25.0
Yes	718	0.05	4.9	42.3
**VAI and BMI**				
Q1–Q3 non-obese	786	0.17	1.2	27.7
Q1–Q3 obese	267	0.08	5.3	51.5
Q4 non-obese	187	−0.04	2.2	35.1
Q4 obese	163	−0.23	9.3	50.0
**VAI and BMI and AO**				
Non-factors	561	0.12	0.9	24.2
Only obese	22	−0.48	0.0	50.0
Only VAI Q4	94	0.16	1.1	27.7
Only AO	225	0.28	1.8	38.2
Obese and VAI Q4	8	−0.30	0.0	100
Obese and AO	245	0.13	5.8	51.7
AO and VAI Q4	93	−0.4	3.4	45.7
All 3 factors	155	−0.23	9.8	33.3

Abbreviations: VAI (visceral adiposity index); BMI (body mass index); AO (abdominal obesity). All comparisons were at a *p* < 0.05 except the ∆ LSMs (BMI and abdominal obesity) and MASLD incidence (VAI).

## Data Availability

Data are contained within the article.

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
