# Peer review of "The Link between Abdominal Obesity Indices and the Progression of Liver Fibrosis: Insights from a Population-Based Study"

_nutrients, 2024, doi:10.3390/nu16111586_

Round 1

Reviewer 1 Report

Comments and Suggestions for Authors

Authors presented interesting paper on association between weight and MASLD. There are some issues that shoud be addressed prior to processing the manuscript

In the conceptualization the use of drugs (especially antidiabetic - this is mentioned in conclusions, hypolipidemic should be assessed). 

Lack of CAP data is puzzling. Was it not assessed?

The precise description of statistical analysis including test used is lacking.  Also, multivariate analysis is not provided in methods.

Figures require addition of statistical significance between groups, while at current form one cannot exclude at which point the difference is significant. Also the manuscript body lacks the “p-values”. Therefore, it is hard to state that the results are meaningful.

Several limitations are pointed out but I am lacking the ethnicity - as far as I understand the data should be limited to population from vicinity of Barcelona (no detailed data is provided)

Comments on the Quality of English Language

Appropriate.

Author Response

We appreciate and are thankful for your suggestions.

Authors presented interesting paper on association between weight and MASLD. There are some issues that shoud be addressed prior to processing the manuscript 

  1. In the conceptualization the use of drugs (especially antidiabetic - this is mentioned in conclusions, hypolipidemic should be assessed).  

The reviewer is right in his/her appraisal. As the reviewer pointed out, one of the limitations of this study is the absence of information regarding chronic pharmacological treatment. While these data are typically documented in the participants' medical records, they were not gathered in this instance. Additionally, it is worth noting that the participants were theoretically healthy individuals from the general population. We have introduced this phrase in limitations: Data on the medication used by the study participants, such as glucose and lipid-lowering drugs, was not available, which could influence the metabolic characteristics of certain patient subgroups, particularly those with dysglycemia”.

  1. Lack of CAP data is puzzling. Was it not assessed? 

We appreciate the reviewer’s comment. Currently, our group possesses a Fibroscan device that is equipped with the CAP algorithm. Regrettably, during the period when the study was conducted, we did not have access to it.

  1. The precise description of statistical analysis including test used is lacking.   

Following the reviewer’s request, in the revised version of the manuscript (Section 2.3) we have  provide a more detailed description about the statistical tests we conducted for data analysis.

  1. Also, multivariate analysis is not provided in methods.

We appreciate the reviewer’s comment. The data regarding the multivariate analysis are already described in Section 2.3 : The relationship between baseline VAI (4th quartile vs 1st-2nd-3rd quartiles), central obesity (BMI≥30 kg/m2), and abdominal obesity, and the prevalence of MASLD (FLI≥60) or hepatic fibrosis (TE≥8 kPa) was assessed using logistic multivariate models. In these models, baseline MASLD or hepatic fibrosis served as the dependent variables, while VAI, central obesity, and abdominal obesity were the main explanatory variables (each assessed separately, not in the same model). The analysis was adjusted for sex, age, and dysglycemia. Additionally, a sensitivity analysis was conducted specifically in individuals with baseline dysglycemia. Nonetheless, we are at the reviewer's disposal should further clarification or information be required.

  1. Figures require addition of statistical significance between groups, while at current form one cannot exclude at which point the difference is significant. Also the manuscript body lacks the “p-values”. Therefore, it is hard to state that the results are meaningful. 

We agree with the reviewer’s comment. Accordingly, statistical tests have been added in the figures 2, 3 and 4, and their significance appears now in the figure notes and in the text body.

  1. Several limitations are pointed out but I am lacking the ethnicity - as far as I understand the data should be limited to population from vicinity of Barcelona (no detailed data is provided)

This is a very interesting comment. The study was conducted in the Northern Metropolitan area of the province of Barcelona, specifically in the regions of Barcelona Nord and Maresme, encompassing participants from 16 primary care centers. Of these, 95% were of Caucasian origin, while 5% represented other ethnicities, predominantly Latin American (3%), with the remaining 2% distributed among African, sub-Saharan, and Eastern ethnicities. The reasons for the lower participation of non-Caucasian subjects were language barriers in understanding the purpose and conduct of the study and difficulty in obtaining administrative contact data due to frequent changes of address in these communities. This data has been added into Results Section: Therefore, the population included in this study was 1,403 subjects (95% were Caucasian) in whom two LSM were obtained, the first at baseline and the second one at the end of follow-up (median 4.2 years later; range, 3.0-5.5).”

Reviewer 2 Report

Comments and Suggestions for Authors

The Authors should analyze the endotoxin behaviour and oxidant stress previously related each other in NAFLD.   Otherwise the role of oxidant stress in the pathogenesis of liver fibrosis is clearly defined. The Authors should dedicate a new chapter to marrkers of oxidant stress, such as serum sp-NOX2 and urinary 8-iso-PGF 2 alpha.

Comments on the Quality of English Language

Moderate editing of English language is required.

Author Response

We appreciate and are thankful for your suggestions.

The Authors should analyze the endotoxin behaviour and oxidant stress previously related to each other in NAFLD.   Otherwise the role of oxidant stress in the pathogenesis of liver fibrosis is clearly defined. The Authors should dedicate a new chapter to markers of oxidant stress, such as serum sp-NOX2 and urinary 8-iso-PGF 2 alpha.

We appreciate the reviewer's suggestions. We agree with the reviewer regarding the role of oxidative stress in the pathogenesis of steatohepatitis and, consequently, progression to fibrosis. However, our present study focused on clinical, anthropometric, and elastographic variables and their association with liver fibrosis progression in the general population. We will take into account your kind suggestions for future studies.

Reviewer 3 Report

Comments and Suggestions for Authors

Link between Abdominal Obesity indices and the Progression of Liver Fibrosis: Insights from a Population-Based Study

Non-alcoholic fatty liver disease (NAFLD) has been considered as one of the most common chronic liver diseases, and the prevalence of NAFLD is increasing all over the word. There is limited research published on the association between VAI and the presence of fibrosis, primarily comprising cross-sectional studies conducted in subjects diagnosed with MASLD, with liver biopsy being the predominant diagnostic method utilized. Data  from previous studies indicate a strong association between different indices of obesity, including Visceral Adiposity Index (VAI), and the presence of  metabolic dysfunction associated steatotic liver disease ( MASLD/MASH), although its relationship with liver fibrosis remains controversial. Introduction provides sufficient background but doesn't include all relevant references regarding NAFLD which must be up-dated. This cross-sectional study protocol was  adhered to the Declaration of Helsinki. Written informed consent was obtained from each patient prior to the study related procedures. Statistical and dosage methods were adequately described and used. Results were include in tables and diagrams.

Author Response

Non-alcoholic fatty liver disease (NAFLD) has been considered as one of the most common chronic liver diseases, and the prevalence of NAFLD is increasing all over the word. There is limited research published on the association between VAI and the presence of fibrosis, primarily comprising cross-sectional studies conducted in subjects diagnosed with MASLD, with liver biopsy being the predominant diagnostic method utilized. Data  from previous studies indicate a strong association between different indices of obesity, including Visceral Adiposity Index (VAI), and the presence of  metabolic dysfunction associated steatotic liver disease ( MASLD/MASH), although its relationship with liver fibrosis remains controversialIntroduction provides sufficient background but doesn't include all relevant references regarding NAFLD which must be up-dated. This cross-sectional study protocol was adhered to the Declaration of Helsinki. Written informed consent was obtained from each patient prior to the study related procedures. Statistical and dosage methods were adequately described and used. Results were included in tables and diagrams.

We appreciate and are thankful for your comment. We have thoroughly reviewed the literature and have added the following bibliographic citations to the introduction. However, if the reviewer believes that any relevant reference on MASLD is absent, please let us know.

Younossi ZM, Golabi P, Paik JM, Henry A, Van Dongen C, Henry L. The global epidemiology of nonalcoholic fatty liver disease (NAFLD) and nonalcoholic steatohepatitis (NASH): a systematic review. Hepatology. 2023 Apr 1;77(4):1335-1347. doi: 10.1097/HEP.0000000000000004. Epub 2023 Jan 3. PMID: 36626630; PMCID: PMC10026948.

Lazarus JV, Mark HE, Anstee QM, Arab JP, Batterham RL, Castera L, Cortez-Pinto H, Crespo J, Cusi K, Dirac MA, Francque S, George J, Hagström H, Huang TT, Ismail MH, Kautz A, Sarin SK, Loomba R, Miller V, Newsome PN, Ninburg M, Ocama P, Ratziu V, Rinella M, Romero D, Romero-Gómez M, Schattenberg JM, Tsochatzis EA, Valenti L, Wong VW, Yilmaz Y, Younossi ZM, Zelber-Sagi S; NAFLD Consensus Consortium. Advancing the global public health agenda for NAFLD: a consensus statement. Nat Rev Gastroenterol Hepatol. 2022 Jan;19(1):60-78. doi: 10.1038/s41575-021-00523-4. Epub 2021 Oct 27. PMID: 34707258.

Rinella ME, Lazarus JV, Ratziu V, Francque SM, Sanyal AJ, Kanwal F, Romero D, Abdelmalek MF, Anstee QM, Arab JP, Arrese M, Bataller R, Beuers U, Boursier J, Bugianesi E, Byrne CD, Narro GEC, Chowdhury A, Cortez-Pinto H, Cryer DR, Cusi K, El-Kassas M, Klein S, Eskridge W, Fan J, Gawrieh S, Guy CD, Harrison SA, Kim SU, Koot BG, Korenjak M, Kowdley KV, Lacaille F, Loomba R, Mitchell-Thain R, Morgan TR, Powell EE, Roden M, Romero-Gómez M, Silva M, Singh SP, Sookoian SC, Spearman CW, Tiniakos D, Valenti L, Vos MB, Wong VW, Xanthakos S, Yilmaz Y, Younossi Z, Hobbs A, Villota-Rivas M, Newsome PN; NAFLD Nomenclature consensus group. A multisociety Delphi consensus statement on new fatty liver disease nomenclature. Ann Hepatol. 2024 Jan-Feb;29(1):101133. doi: 10.1016/j.aohep.2023.101133. Epub 2023 Jun 24. PMID: 37364816.

Cusi K, Isaacs S, Barb D, Basu R, Caprio S, Garvey WT, Kashyap S, Mechanick JI, Mouzaki M, Nadolsky K, Rinella ME, Vos MB, Younossi Z. American Association of Clinical Endocrinology Clinical Practice Guideline for the Diagnosis and Management of Nonalcoholic Fatty Liver Disease in Primary Care and Endocrinology Clinical Settings: Co-Sponsored by the American Association for the Study of Liver Diseases (AASLD). Endocr Pract. 2022 May;28(5):528-562. doi: 10.1016/j.eprac.2022.03.010. PMID: 35569886.

Stefan N, Cusi K. A global view of the interplay between non-alcoholic fatty liver disease and diabetes. Lancet Diabetes Endocrinol. 2022 Apr;10(4):284-296. doi: 10.1016/S2213-8587(22)00003-1. Epub 2022 Feb 17. PMID: 35183303.

Round 2

Reviewer 1 Report

Comments and Suggestions for Authors

Authors responded satisfactory to the issues raised.

Author Response

Dear reviewer,

Thank you for your comments.

We are sending you the final version of manuscript with the reviewed changes.  In this latest version,  changes suggested by the academic editor are highlighted in yellow.

Reviewer 2 Report

Comments and Suggestions for Authors

The Authors answered correctly to all my queries

Comments on the Quality of English Language

Moderate editing of English language is required

Author Response

(The authors gave the same response as above.)
